# Molecular Detection and Associated Risk Factors of *Brucella melitensis* in Aborted Sheep and Goats in Duhok Province, Iraq

**DOI:** 10.3390/pathogens12040544

**Published:** 2023-04-01

**Authors:** Alind M. Ameen, Nadhim S. Abdulaziz, Nacheervan M. Ghaffar

**Affiliations:** 1Duhok Veterinary Directorate, Ministry of Agriculture and Water Resources, Duhok 42001, Iraq; 2University of Zakho, Zakho International Road, Duhok 42002, Iraq; 3College of Veterinary Medicine, University of Duhok, Zakho Road, Zakho 42001, Iraq

**Keywords:** sheep and goats, *B. melitensis*, risk factor, RT-PCR

## Abstract

Brucellosis in sheep and goats has a significant economic and zoonotic impact on the livestock population of Duhok province, Iraq. A total of 681 blood samples from aborted sheep and goats were collected from different flocks in seven districts of Duhok and tested using real-time polymerase chain reaction (RT-PCR). Logistic regression was used for the analysis of the potential risk factors associated with RT-PCR positivity. Results revealed an overall prevalence of 35.45% (CI = 2.57) and 23.8% 18 (CI = 0.44) in sheep and goats, respectively. A statistically significant (*p* = 0.004) difference in prevalence was found between the two species. RT-PCR detected more positive cases in older-aged animals (OR = 0.7164; *p* = 0.073). A significant difference was found in RT-PCR positivity in relation to different risk factors, including body condition, treatment taken, and abortion frequency (<0.001). The phylogenetic tree based on the 16S rRNA gene indicated that the isolates belonged to *B. melitensis* and shared a common ancestor and were genetically related to the United States of America (USA), Greece, China, and Nigeria. This study demonstrates that brucellosis is widely prevalent in the study regions. Therefore, the study suggests the implementation of preventive control measures for brucellosis.

## 1. Introduction

Brucellosis is one of the most important and common bacterial zoonoses. It is widespread worldwide and has major economic and public health significance [1]. Brucellosis causes substantial losses in domestic animals, mainly sheep, goats, and cows. It is a major zoonotic disease that can cause reproductive disorders in animals. The most common *Brucella* spp. that cause brucellosis are *Brucella abortus* (*B. abortus*) in cattle, *Brucella melitensis* (*B. melitensis*) or *Brucella ovis* (*B. ovis*) in small ruminants, *Brucella suis* (*B. suis*) in pigs, and *Brucella canis* (*B. canis*) in dogs [2,3,4]. These species are Gram-negative coccobacilli, facultative intracellular, non-motile, and non-spore forming bacteria. Currently, with a preference for different hosts, twelve Brucella species have been reported [5]. Brucella is a highly infectious bacterium, as even 10–100 cells are sufficient to cause systemic infection [6,7].

According to the World Health Organization (WHO), Food and Agriculture Organization (FAO), and World Organization for Animal Health (WOAH), brucellosis is still one of the most important and prevalent zoonoses worldwide. Brucellosis is also classified as one of the seven neglected diseases, due lacking a distinctive clinical presentation in humans [8].

Brucellosis has an important impact on animal industries and human health worldwide. Although brucellosis has been eliminated in many developed countries based on proper prevention and eradication, it remains endemic in large regions, mainly in Africa, the Middle East (including Iraq), the Mediterranean Basin, and the Indian subcontinent. In addition, in some developed countries with limited resources, low income, and frequent contact with livestock animals, Brucellosis is still common. Almost 500,000 new brucellosis cases are reported yearly in humans globally, with around 10/100,000 population. Nevertheless, this figure might underestimate the epidemiology and the actual number, which is proposed to be 5,000,000 to 12,500,000 cases/year [9,10]. Due to abortion, Brucella infection is considered as one of the main factors that can cause massive economic losses in sheep and goats. Aborted animals due to Brucella infection commonly occur during the last third of pregnancy, in addition to stillbirths, weak lambs and kids, and infertility, which may be another complication of this bacterium [7]. In males, it is characterized mainly by epididymitis and orchitis, while, in females, it is characterized by placentitis in pregnant animals, with excretion of the organisms in milk and uterine discharges in female animals [7]. The massive economic losses due to brucellosis in sheep and goats mandate the use of rapid and sensitive diagnostic techniques for appropriate detection, so as to determine and implement proper control strategies for the eradication of this disease. However, some of the diagnostic techniques are not fully suitable, taking a long time, being labor-intensive, and not being sensitive in chronic cases [11,12]. Bacterial isolation is also time-consuming and risky due to the high possibility of laboratory-acquired infection [11,12]. Therefore, real-time PCR (probe-based method) was used in this study as a rapid, fast, and more sensitive tool over other diagnostic methods including conventional PCR. This technique’s results can be evaluated without gel electrophoresis, which can help to reduce the experimental time and increase throughput [12]. RT-PCR is applied in clinical microbiology and is not only limited to detecting and identifying the targeted organism, but also enables use for the characterization of other targets, such as antimicrobial resistance and virulence genes [13]. It is an effective method to detect slow-growing or non-culturable organisms. The intracellular location of brucella in such tissue as reticuloendothelial cells and bone marrow limits the effectiveness of antibiotics. An antimicrobial course with quinolones, doxycycline, rifampicin, streptomycin, and aminoglycoside, alone or in combination, is used to treat brucellosis [14]. The detection of abortigenic agents rapidly, at the early stage of an outbreak, using molecular techniques, helps in minimizing the spread of infection and the outbreak probability and increasing the treatment efficiency. Thus, RT-PCR is increasingly used as a diagnostic tool for the etiologic diagnosis of abortion in cattle [15,16]. Seroprevalence studies using both Rose Bengal and ELISA tests were conducted in different northern governorates in Iraq, including Duhok, and determined that brucellosis is endemic in Iraq and poses an important zoonotic risk to the human community [17,18,19].

In the Kurdistan region, Iraq, the consumption of milk from various animals, including sheep, goats, cows, and buffaloes, has been reported as the main route of transmission and infection in humans. Recently, it has been reported that the incidence of human brucellosis in the Kurdistan region is high compared with the incidence in neighboring countries and it remains a challenging health problem [20,21]. This might be due to having borders with Iran, Turkey, and Syria; the presence of conflicts; the lack of preventive and control measures, and illegal animal movement through uncontrolled border ports, increasing the risk of brucellosis [22]. Some seroprevalence studies on brucellosis in humans and animals have been conducted in the Middle East, which has been considered as an endemic area [22]. In fact, the highest incidence of human brucellosis has been reported in five–ten countries in this area, including Syria, which has the highest incidence of brucellosis worldwide [22].

Therefore, the present study aimed to detect and identify *B. melitensis*, one of the critical abortion agents, from blood samples collected from aborted sheep and goats in the Duhok governorate using RT-PCR and to identify the associated risk factors from the RT-PCR-positive cases.

## 2. Materials and Methods

### 2.1. Study Area Description

The study was conducted using samples from all districts of the Duhok governorate, Kurdistan region, Iraq. The city lies in the northwest of Iraq and the western part of the Kurdistan region, around 470 km north of Baghdad and 430–450 m above the sea level, and it borders Syria and Turkey. The area is presently populated by approximately 2 million people and contains around 1 million sheep and goats [18]. Along with another six districts, Sumeil, Zakho, Amedi, Shekhan, Bardarash, and Akre (Figure 1), Duhok covers 10,715 square km and lies at latitude 36 north and longitude 43 east. The agroecology of this governorate is characterized by a hot summer Mediterranean climate (Csa), showing sweltering, virtually rainless summers and cool to cold, wet winters. Precipitation falls in the cooler months, being heaviest in late winter and early spring. The city can receive around two or three snowy days yearly, with more severe snowfall in the uplands. Summers are virtually rainless, with rain returning in late autumn. The geographical situation of this province and the movement of the animals across the borders in this area might contribute to the increased prevalence of different diseases, including brucellosis. This may affect the sheep and goat sectors, which are important in sustaining the country’s food security.

### 2.2. Data Collection

From each herd that had abortion cases, the following information was collected: location, source of animals, number of heads, number of aborted ewes and does, time of abortion, clinical features of dams, clinical signs of aborted fetus, history of herd abortion, and treatment. These were recorded separately on sample data sheets.

### 2.3. Samples Collection

Blood samples were collected aseptically from aborted sheep, *n* = 488, and goats, *n* = 193 (*n* = 681), having clinical signs of brucellosis, from different herds in Duhok province (Duhok, Akre, Zakho, Amedi, Shekhan, Sumeil, and Bardarash districts). The main clinical manifestations in these animals were reproductive failures, a large number of abortions, and the birth of weak offspring. Generally, brucellosis causes abortion during the last two months of pregnancy. Blood (5 mL) was collected from the jugular veins of the aborted small ruminants using disposable needles (18 gauges) and 10 mL syringes. EDTA tubes were used to keep the DNA integrated during storage and transportation. Blood samples were separated in accordance with the regions from which samples were collected. All samples were sent to the laboratory in ice boxes (4 °C). DNA extraction from all blood samples was performed in the laboratory as soon as the samples arrived.

### 2.4. Extraction of Genomic DNA

Genomic DNA was extracted from collected samples by using a DNA purification kit (GeneAll, Seoul, Republic of Korea). Briefly, 20 µL of Proteinase K was added to the bottom of a 1.5 mL microcentrifuge tube and then 200 µL of blood was transferred to the tube and it was incubated for 2 min at room temperature. After this, 200 µL of buffer (BL) was added to the tube, mixed thoroughly by vertexing, and incubated at 56 °C for 10 min. Later, 200 µL of absolute ethanol was added to the sample, and it was vortexed and spun down briefly. The mix was transferred to an SV silica membrane column carefully, centrifuged for 1 min at 8000 rpm, and then the collection tube was replaced. This was followed by adding 600 µL of buffer and it was centrifuged for 1 min at 8000 rpm and the pass-through was discarded. The tube was then centrifuged at full speed for 1 min in a fresh 1.5 microcentrifuge tube. Finally, 100–200 µL of buffer AE was added to the center of the tube for the optimal elution of DNA and it was incubated for 1 min at room temperature, and then centrifuged at full speed for 1 min. The concentration and purity of the extracted DNA were checked using a Nanodrop (Thermofisher, Waltham, MA, USA). The obtained DNA was stored at −20 °C to be used for conventional and real-time PCR.

### 2.5. Real-Time PCR Amplification

The extracted genomic DNA was subjected to RT-PCR for the amplification of *Brucella* spp. One-step PCR kits provide components for “one-step” RT-PCR detection in a convenient format that is compatible with both rapid and standard qPCR cycling conditions. The RT-PCR assay was performed in a total reaction volume of 15 μL consisting of 7.5 μL of universal qPCR Master Mix (Bioingentech, Concepción, Chille), 0.3 μL of Primer, Probes, and Internal Control Universal Mix (Bioingentech, Concepción, Chille), 3.7 μL of PCR-grade water, and 3.5 μL of extracted DNA as a template (50 ng/μL). The amplification and fluorescence detection were performed on a Step One Plus RT-PCR system (Agilent, Malaysia) using the thermal conditions as follows: preheating at 50 °C for 2 min, initial denaturation at 95 °C for 2 min, 40 cycles of denaturation at 95 °C for 30 s, annealing at 58 °C for 30 s, and extension at 72 °C for 30 s.

### 2.6. PCR Amplification of 16S rRNA and Sequence Analysis

The extracted DNA was also used in conventional PCR (Applied Biosystems, Waltham, MA, USA) for 16S rRNA gene amplification of *Brucella* spp. universal primers (Table 1).

The PCR assay was performed in a total reaction volume of 40 μL consisting of 20 μL of 2X Master Mix (JenaBiosciences, Jena, Germany), 2 μL of forward and 2 μL reverse primers (10 pmol/μL), 10 μL of extracted DNA as a template (50 ng/μL), and 6 μL of deionized nuclease-free water. The cyclic conditions used in PCR were as follows: initial denaturation at 94 °C for 5 min, 35 cycles of denaturation at 94 °C for 45 s, annealing at 50 °C for 60 s, and extension at 72 °C for 90 s, followed by final extension at 72 °C for 10 min. The PCR products were run on 1% agarose gel, stained with safe red dye (JenaBiosciences, Jena, Germany), and visualized under a gel documentation system (Bio-Rad, Hercules, CA, USA).

### 2.7. Sequencing of the 16S rRNA Gene and Phylogenic Analysis

The PCR products for 16S rRNA from *B. melitensis* were subjected to single end sequencing using a reverse primer, via the dideoxy chain termination method (Macrogen, Seoul, Republic of Korea). The quality of these sequences was checked, and they were cleared of noise. Cleared sequences from different geographical areas were submitted to the NCBI and deposited in GenBank; the accession numbers of submitted samples are shown in Table 2.

The phylogenic tree was produced using the 16S rRNA sequences of seven samples from this study and compared with another 19 *Brucella* spp. from a database selected based on the isolation source, host, and geographical area. These sequences were obtained from GenBank, and multiple alignments with the Clustal W method by MEGA11 were implemented to perform a phylogenetic analysis using neighbor joining. The evolutionary distances were computed using the Jukes–Cantor method. The bootstrap measures were determined from 1000 repeats of the original data.

### 2.8. Data Management and Analysis

The data that were obtained from this study were recorded, coded, and stored in Microsoft Excel for Windows 11 and transferred to GenStat Release 12.1. Molecular screening results as a prevalence rate were calculated from the total number of aborted sheep and goats sampled and positive samples. The association between brucellosis and presumptive risk factors was analyzed using logistic regression analysis. The variables with a *p*-value less than or equal to 0.25 in univariable logistic regression, after checking for multicollinearity using the collinear matrix index and an interaction effect using cross-product terms, were taken forward for multivariable modeling. For all statistical analyses, confidence intervals (CI) of 95% and *p*-values of 0.05 were used.

### 2.9. Ethical Approval

The study was approved by the Ethical Committee of the College of Veterinary Medicine, Duhok University, to ensure that the experiment was performed according to their regulations and supervision (permit number: CVM2021/220UD).

## 3. Results

This study was conducted using blood samples from seven different Duhok districts, collected from 681 aborted sheep and goats. Table 3 illustrates the details of the prevalence rates irrespective of district, breed, and abortion rate.

In this study, a total of 681 blood samples from aborted sheep and goats were collected in seven different Duhok districts and molecularly screened for *B. melitensis* using RT-PCR. From these samples, 32.16% (*n* = 219) were positive for *B. melitensis* and these included 35.45% (173/488) of sheep and 23.8% (46/193) of goats, and there were statistically significant differences (*p* < 0.004).

Regarding the abortion cases, the data illustrated that there was a significant difference between Duhok districts (*p* = 0.028). In addition, there was a significant difference in abortion cases due to these bacteria from Zakho compared with the Akre district, with the odds ratio of molecular screening being 2–3-fold higher in aborted animals in the Zakho district than the Akre district (*p* < 0.001), while there were no significant differences with other districts—Sumeil, Duhok, Bardarash, and Shekhan.

Related to the abortion cases in different age groups, molecular screening for *B. melitensis* in this study found no statistically significant differences among the infected animals’ age groups (*p* = 0.167). Although there were more positive cases detected in older age groups (OR = 0.7164, *p* = 0.073) than the youngest age groups (OR = 0.7699, *p* = 0.244), there were no significant differences observed. Concerning the molecular screening of *B. melitensis* among the goat breeds, the native breed was approximately 1.4 times more positive for this bacterium when compared with the Afghani breed (OR = 0.7089; *p* = 0.642).

Based on the molecular screening of *B. melitensis*, there was a significant difference among sheep and goats with a history of abortion and the healthy group compared with those animals that had a history of abortion and sickness (*p* < 0.001). The aborted and healthy animals were almost five-fold less frequently infected with this bacterium than previously infected animals (OR = 5.3, *p* < 0.001). It is interesting to note that the sheep and goats that received treatment were almost less likely to be infected with this bacterium when compared with those animals that did not receive any previous treatment (OR = 0.4551, *p* < 0.001; Chi2 < 0.001). In addition, the frequency of aborted sheep and goats that experienced abortion one time was higher than those with multiple abortions, and this result showed a statistically significant difference (*p* < 0.001) (*p* < 0.001) (Table 4). Furthermore, sheep and goats from large flock sizes were also found to be at a higher risk of Brucella infection than those from small flock sizes (*p* = 0.019 and OR = 1.477).

Although the test used in this study was extremely sensitive (probe-based method), mixed infection with the other abortion-causative agent was challenging. Therefore, RT-PCR kits to screen another common bacterium that might cause abortion (data not shown) in small ruminants were used. The study found mixed infection with other abortogenic causative agents (Figure 2). However, only single-infection samples were sent for sequencing and deposited in GenBank as uncultured brucella samples to validate the RT-PCR results and compare them with the other data submitted to NCBI using the phylogenetic tree.

The phylogenetic tree, based on the 16S rRNA gene, showed and confirmed that the isolated samples from seven Duhok districts belonged to *B. melitensis* (Figure 3). The phylogenetic tree revealed that all these isolates were gathered in one common ancestor and had a genetic relationship with other geographical areas and different isolation sources in different countries.

The partial sequences of 16S rRNA from different isolation sources were compared with 19 sequences of *Brucella* spp. that were retrieved from GenBank. These sequences were selected based on the isolation sources and geographical areas, i.e., different countries. The phylogenetic tree of these sequences was distinctly divided into two main clusters, named group A and B, which covered two *Brucella* spp. (*B. melitensis* and *B. abortus*); the samples of this study were represented in the main cluster named group A, which was grouped with the samples of different isolation sources from different countries. In general, group A was subdivided into two sub-groups. Figure 2 shows the topology of the phylogenetic tree, and the isolates of this study were distributed between these two groups. Briefly, within the first sub-group, the tree shows that the sample from Zakho (OP363358) is genetically similar to that of Akra (OP363357), and they are more genetically related to samples isolated from China (MT611103) as they shared a common ancestor. The same pattern can be seen between the Shekhan and Amedi samples, while Shekhan (OP363356) is more genetically close to the brucella sample isolated from camels in Saudi Arabia (MN235870). However, within the second sub-group of group A, the samples taken from aborted sheep in Duhok, Sumeil, and Bardarash are similar and grouped together; however, the Bardarash sample (OP363359) is more genetically related to the *Brucella* spp. isolated from female and male marine toads, in the USA (MT471348). However, in group B, the current study’s isolates were clustered and grouped with other countries’ isolates.

The evolutionary history was inferred using the neighbor-joining method. The percentage of replicate trees in which the associated taxa clustered together in the bootstrap test (1000 replicates) is shown next to the branches. The tree is drawn to scale, with branch lengths in the same units as those of the evolutionary distances used to infer the phylogenetic tree. The evolutionary distances were computed using the Jukes–Cantor method and are given in the unit of the number of base substitutions per site. Evolutionary analyses were conducted in MEGA11.4.

## 4. Discussion

The current study presented significant insights into the most important bacterial causative agent that leads to reproductive failure in sheep and goats in different Duhok governorate districts. The study demonstrated the prevalence of *B. melitensis* as a causative agent of reproductive failure and the role of the possible risk factors in this disease, such as the geographical area, age, breed, body condition, treatment, etc., in sheep flocks (Table 4).

This work is considered as the first molecular screening study conducted for *B. melitensis* in aborted sheep and goats and offers a good epidemiological understanding that might be useful to improve the management activities in the study areas and provide insights for veterinary services to implement better preventive measures to control this issue. A serological study was conducted in the same area by [18]. Using the Rose Bengal test (RBT) and indirect ELISA (iELISA), they reported that 31.7% of sheep and 34.0% of goats were positive for Brucella. However, as they stated in their study, there were a few limitations related to the tests used, area covered, and the period of the study due to unintended reasons. Therefore, this study tried to address all the limitations mentioned before using RT-PCR. Interestingly, equivalent results were reported using different methodologies.

Multivariable model analysis was used to identify different risk factors associated with brucella infection in small ruminants using RT-PCR. Generally, this study found that there was a significant difference in Brucella-positive samples among the Duhok districts. These findings are in line with a serological study in the same governorate by [18], where the researcher found that the seropositivity of this bacterium was significantly higher among animals sampled from three districts: Akre, Zakho, and Sumeil [18]. In contrast, our result disagreed with another serological study that was conducted in the Duhok governorate that reported lower seropositivity results in both sheep and goats [24]. In Iraq, another study reported a higher prevalence rate of 59.5% [25], and lower results were reported by [19,26]. The differences in these observations could be due to different factors, including differences in the agroecological areas of these studies, the management and production systems, and the sample size, as well as the various diagnostic tests used [27].

It is not surprising that a highest prevalence rate of RT-PCR-positive samples of 43.6% was recorded in the Zakho district, followed by the Duhok and Sumeil districts. The geographical situation of this district and the movement of animals across the Iraqi border in this area might contribute to the increased prevalence of different diseases, including brucellosis. In addition, the higher prevalence rate of brucellosis in sheep in the Zakho district might be also related to poor management, as well as environmental factors.

The Zakho district, having one of the main ports in Iraq, is considered as the major route for the import of animals and animal products to different governorates of Iraq. This might increase the likelihood of contact between local animals and others, especially as there is no proper risk assessment and risk analysis conducted on the imported animals. Thus, it might serve to increase the risk of transmission of different diseases, including Brucella. Moreover, the districts with higher brucellosis RT-PCR positivity share the most borders with Turkey, Syria, and the Mosul province, where most of the uncontrolled movement of animals occurs; the uncontrolled movement and smuggling of animals across borders could contribute to the spread and persistence of animal diseases in a regional context [18]. In addition to this factor, imported small ruminants with an unidentified history from neighboring countries are considered a significant risk factor of seropositivity for Brucella in small ruminants [28].

Although this study reported more positivity in older than younger groups, no significant difference was observed among sheep and goat age groups. This finding is in disagreement with two previous African studies, as researchers indicated that age is considered one of the fundamental factors that influence brucellosis positivity in Zambia [29], and in Southern and Eastern Ethiopia [30]. This variation may be due to previous infection, vaccination, or poor management systems.

This could be attributed to the fact that reproductively active animals are more susceptible to Brucella infection than sexually immature animals, as the sex hormones that promote the growth and multiplication of Brucella organisms likely increase in concentration with age and sexual maturity [31]. This result may be due to the continuous exposure of animals over time or the fact that the growth-stimulating factor becomes more abundant in sexually mature animals [32]. The differences in the results between the current and previous studies may be the result of ongoing vaccination control programs of lambs and kids from 3 to 6 months of age in the Kurdistan region.

In terms of animal species, this study found a significant difference among sheep and goats’ positive samples, while no significant difference was noticed among sheep and goat breeds. Keeping sheep in contact with Brucella-infected goats is also considered as a potential risk factor for brucellosis spread among sheep flocks [33]. This result is not in conflict with some earlier studies conducted in the Duhok province [19,24,34]. Keeping herds with multiple livestock species leads to higher seropositivity for Brucella infection, indicating opportunities for cross-species infection. In addition, keeping multiple livestock species and the herding of small ruminants besides cattle or camels have also been stated as risk factors for Brucella infection [23].

The current study showed that the abortion frequencies in sheep and goats affected once were higher than those that experienced abortion twice or more. This is an indication of the immune response from the previous abortion due to acquired or specific immunity developed after exposure to the same antigens [35].

Moreover, the current study results showed that the flock size is another risk factor, in which higher RT-PCT positivity was reported in sheep and goats from large flocks and medium flocks than in those from small flocks. These results are in accordance with previous studies in cattle in Southern and Eastern Ethiopia [36,37,38] and might be supported by the fact that an increase in flock size is normally associated with increasing flock density, as one of the contributing factors to exposure to Brucella infection mainly after abortion [36,39]. The herding of multiple livestock species together as one large and high-density flock, mainly keeping goats and sheep along with cattle, has been revealed as an essential determinant risk factor of Brucella seropositivity [40].

Previously treated animals with some drugs, i.e., sheep and goats that received treatment, were almost less likely to be re-infected with Brucella organisms than those that did not receive treatment. This increases the responsibility of veterinary services to provide a proper diagnosis and treatment for infected animals, to reduce the possibility of re-infection. However, this study identified mixed infection with other abortogenic bacterial causative agents; therefore, it is recommended for veterinary services to implement proper preventive and control measures for brucellosis considering the treatment of the mixed infection issue.

In this context, the phylogenetic tree result of our study found a close genetic relationship between Zakho samples and those from China, but more identical to samples taken from Akre. The same types of relations were found between the Shekhan and Amedi samples, but Shekhan was more genetically related to the *Brucella* spp. sample isolated from camels in Saudi Arabia. Nevertheless, the samples taken from aborted sheep in Duhok, Sumeil, and Bardarash were genetically similar; however, the Bardarash sample was more genetically related to the *Brucella* spp. isolated from female and male marine toads, in the USA. This result emphasizes the role of animal and animal product importation without proper risk analysis in formal and informal border pots. Another possible reason for this finding might be due to the sharing of male animals in the breeding season in this area.

## 5. Conclusions

Brucellosis is highly prevalent in this area, mainly at the herd level, and can be considered an economic and public health concern. This study identified the main risk factors for brucellosis positivity by RT-PCR in sheep and goats in different Duhok governorate districts. These results showed that the district, species, history of abortion, previously treated animals, and frequency of abortion were highly associated with brucellosis in sheep and goats in this area. The sequenced samples taken from different districts were more genetically related to the data retrieved from different sources of isolates in China and the USA. This gives an indication that animals originating from outside the herd are an important source of Brucella infection. Therefore, a strict control procedure for brucellosis in Iraq should be enforced, and proper measures for animal transportation, mainly illegal importation, need to be implemented. Despite the presence of mixed infection in some collected samples with other causative agents of abortion, these findings provide some indication of the proper and appropriate, sensitive tests used to study brucellosis in this area. These results can be considered as a basis for undertaking a broad study focusing on the best control program and best treatment for this disease in this area.

## Figures and Tables

**Figure 1 pathogens-12-00544-f001:**
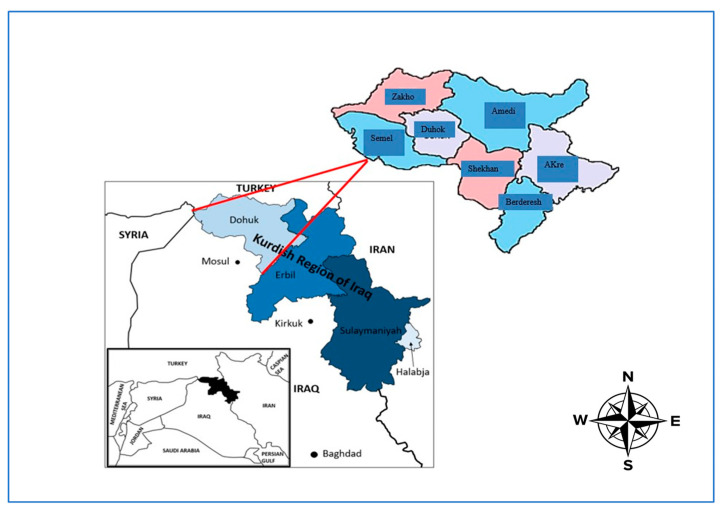
Map of study area, showing seven targeted districts in Duhok governorate, Kurdistan region of Iraq.

**Figure 2 pathogens-12-00544-f002:**
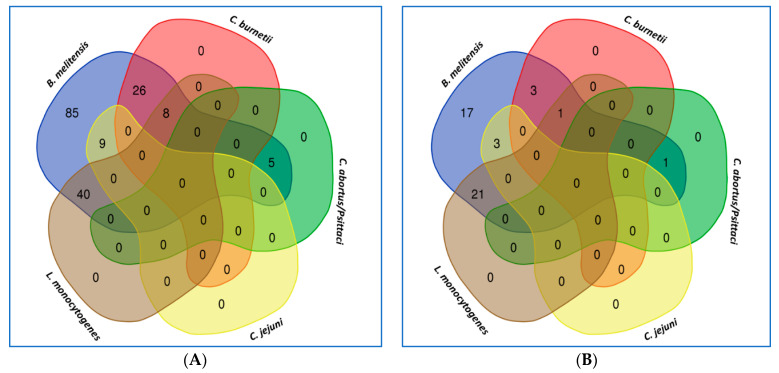
Illustration of the mixed infection of *B. melitensis* with other abortogenic causative agents in aborted cases in sheep (**A**) and goats (**B**). The Venn diagram was obtained online using (http://bioinformatics.psb.ugent.be/webtools/Venn/) (accessed on 15 February 2023).

**Figure 3 pathogens-12-00544-f003:**
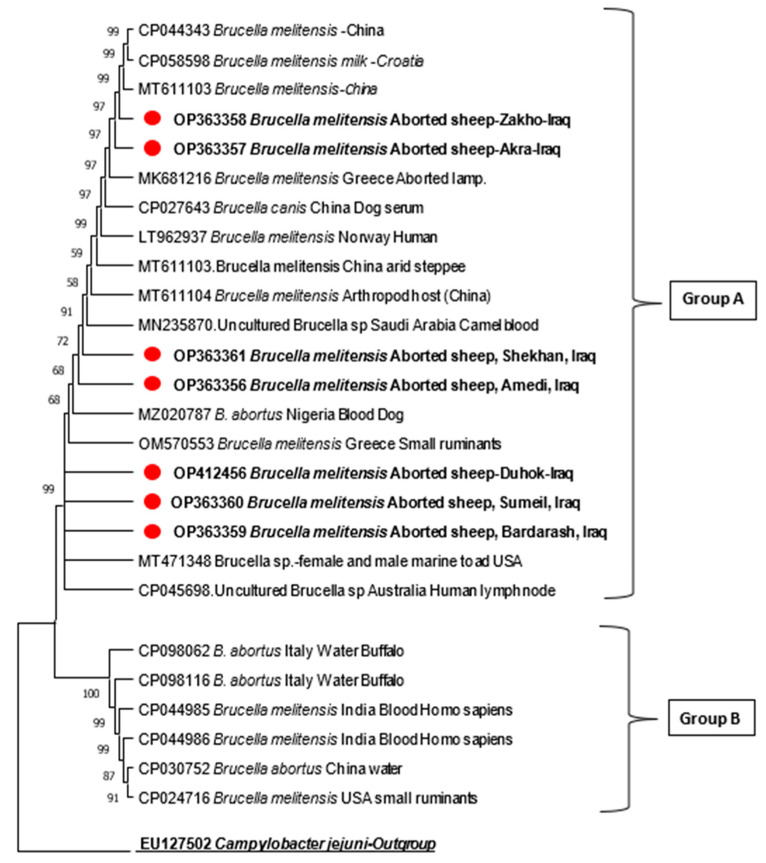
The 16S rRNA-based phylogenetic tree of *B. melitensis* using 7 partial sequences from this study compared with 19 respective sequences available in database. This tree is divided into two main clusters, named group A and B; this study’s isolates were clustered with group A.

**Table 1 pathogens-12-00544-t001:** The 16S rRNA primers used for conventional PCR in this study.

Primer Name	Gene	Sequence (5′-3′)	Amplicon Size (bp)	References
515f	16S rRNA	GTGCCAGCMGCCGCGGTAA	600	[23]
806r		GGACTACHVGGGTWTCTAAT		

**Table 2 pathogens-12-00544-t002:** Accession numbers of deposited samples in GenBank.

SPP	Accession No.	District
*B. melitensis*	OP363356	Amedi
*B. melitensis*	OP363357	AKRA
*B. melitensis*	OP363358	Zakho
*B. melitensis*	OP363359	Bardarash
*B. melitensis*	OP363360	Sumeil
*B. melitensis*	OP363361	Shekhan
*B. melitensis*	OP412456	Duhok

**Table 3 pathogens-12-00544-t003:** Prevalence rates of abortion among sheep and goat breeds per district using real-time PCR.

District	Animal Species						Total
	Sheep			Goat			
	Karadi (%)	Awasi (%)	Hamdani (%)	Native (%)	Shami (%)	Afghani (%)	
Sumeil	9/28 (32.14%)	10/33 (30.3%)	11/25 (44%)	7/18 (38.88%)	1/10 (10%)	0/1 (0%)	38/115 33.04%
Zakho	10/24 (41.66%)	12/29 (41.37%)	18/31 (58.06%)	3/13 (23.07%)	4/11 (36.36%)	1/2 (50%)	48/110 43.63%
Duhok	8/23 (34.78%)	13/26 (50%)	13/30 (43.33%)	4/15 (26.66%)	2/11 (18.18%)	0/0 (0%)	40/105 38.09%
Bardarash	3/8 (37.5%)	1/9 (11.11%)	3/11 (27.27%)	2/10 (20%)	1/8 (12.5%)	0/0 (0%)	10/046 21.73%
Amedi	8/20 (40%)	5/23 (21.7%)	5/19 (26.3%)	4/18 (22.22%)	5/17 (29.41%)	2/3 (66.66%)	29/100 29%
Shekhan	7/21 (33.33%)	8/22 (36.36%)	4/16 (25%)	2/13 (15.38%)	1/8 (12.5%)	0/0 (0%)	22/80 27.50%
Akre	6/26 (23.07%)	6/29 (20.68%)	13/35 (37.14%)	6/20 (30%)	1/12 (8.33%)	0/3 (0%)	32/125 25.60%
Total	51/150 (34%)	55/171 (32.16%)	67/167 (40.1%)	28/107 (26.2%)	15/77 (19.5%)	3/9 (33.33%)	

**Table 4 pathogens-12-00544-t004:** Risk factors associated with abortions using univariable Poisson regression analyses.

Risk Factor	Category	Total	Positive Sample	Positive (%)	*p* Value	CI 95%		Odds Ratio	Chi2
						Lower CI	Upper CI		
District	Sumeil	115	38	33.04	0.206	0.8202	2.5	1.43	0.028
	Zakho	110	48	43.63	0.004	1.297	3.9	2.25	
	Duhok	105	40	38.09	0.043	1.019	3.13	1.78	
	Bardarash	46	10	21.73	0.603	0.36	1.81	0.80	
	Amedi	100	29	29	0.569	0.6581	2.14	1.187	
	Akre	125	32	25.6	<0.001 Ref	0.2303	0.51	0.34	
	Shekhan	80	22	27.5	0.763	0.5847	2.07	1.1	
Age	≤1	130	39	30	0.244	0.4958	1.19	0.76	0.167
	2–3	316	113	35.75	<0.001 Ref	0.4423	0.7	0.55	
	≥4	235	67	28.5	0.073	0.4974	1.03	0.71	
Species	Sheep	488	173	35.45	0.004	1.201	2.56	1.75	<0.004
	Goat	193	46	23.8	<0.001 Ref	0.2247	0.43	0.31	
Breed (Sheep)	Karadi	150	51	34	0.727	0.6818	1.73	1.08	0.282
	Awasi	171	55	32.16	<0.001 Ref	0.344	0.65	0.47	
	Hamdani	167	67	40.1	0.128	0.9048	2.20	1.4	
Breed (Goat)	Native	107	28	26.16	0.642	0.166	3.02	0.70	0.46
	Shami	77	15	19.48	0.342	0.1084	2.16	0.48	
	Afghani	9	3	33.33	0.327 Ref	0.125	1.99	0.5	
Body condition	Sick	251	104	41.43	<0.001	1.394	2.69	1.93	<0.001
	Healthy	430	115	26.74	<0.001 Ref	0.2949	0.45	0.36	
Treatment taken	Treatment	251	55	21.9	<0.001	0.3186	0.65	0.45	<0.001
	No treatment	430	164	38.13	<0.001 Ref	0.5075	0.74	0.61	
Abortion frequency	Once	342	138	40.35	<0.001	1.666	3.46	2.40	<0.001
	2	255	56	21.9	<0.001 Ref	0.2092	0.37	0.28	
	>2	84	25	29.76	0.147	0.8655	2.62	1.50	
Flock size	Small (<10)	246	67	27.23	0.536	0.5751	1.33	0.87	0.019
	Medium (10–15)	248	96	38.7	0.058	0.9864	2.21	1.47	
	Large (>15)	187	56	29.9	<0.001 Ref	0.3126	0.58	0.42	

## Data Availability

Raw data for sequences are available in NCBI (accession numbers provided). The ethic approval letter for CVM2021/220UD will be provided upon request.

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
