# Peer review of "Molecular Detection and Associated Risk Factors of Brucella melitensis in Aborted Sheep and Goats in Duhok Province, Iraq"

_pathogens, 2023, doi:10.3390/pathogens12040544_

Round 1

Reviewer 1 Report

The manuscript discusses molecular detection and associated risk factors analysis of Brucella melitensis (B. melitensis) in aborted Sheep and Goats, in different districts of Duhok province. It needs major modifications as below:

 In the title, the name of Iraq should be added.

Body condition, Treatment taken should be written in small letters.

The phylogenetic tree based on 16S rRNA gene  genetically related to USA, Greece, China, Nigeria, and other countries .. other countries should be deleted and the most similar countries should be written in this section.

In the text, Brucellosis should be written with a small letters.

The prevalence of brucellosis in Duhok province and other parts of Iraq should be introduced in the introductions and then the importance of this region for brucellosis investigation in small ruminates describe.

Line 88, some seroprevalence studies on brucellosis in human and animals need accurate references.

In the Study area Description, the importance of Duhok province for brucellosis study in sheep and goats and the criteria for selection of this region describe.

Line 116, in the sample collection, you should identify the number of  aborted sheep and goats separately.

Line 126, you told that DNA extraction from all blood samples was done in the laboratory as soon as the samples arrived, how many samples did you extract in one day? Why you did not preserve the samples at -20°C?

Line 147, what is the target gene for Real-time PCR amplification? Include the primers and band size for Real time PCR

Line 173, what is your mean about The PCR products for 16Sr RNA from B. melitensis? Did you isolate the bacteria from the blood? How did you identify the B. melitensis?

Line 178, as the sampling is from Blood, what is the meaning of aborted sheep in isolation source of table 2?

Line 161, PCR amplification of 16S rRNA and sequence analysis did for all samples or only for positive samples in Rt-PCR? Include in the method section.

The resolution of figure 2 needs to improve. Group a and b should be described in the legend.

Describe groups A and B clearly in the text. What is the difference between groups A and B.

The clear results of the phylogenetic analysis should be clearly reported in the result section, this tree is unclear and ambiguous.  

Figure 3 in the discussion is not necessary and should be deleted. The results and risk factors of this study should be discussed clearly. 

Author Response

Dear Reviwer

Manuscript ID pathogens-2277585 entitled " Molecular detection and associated risk factors analysis of Brucella melitensis (B. melitensis) in aborted Sheep and Goats, in different districts of Duhok province-Iraq ".

We thank yourself  for the put into consideration of our submission and we are very happy to revise the manuscript in line with the recommendations, inputs, edits, and feedback provided.

We indicate below the changes made in text background in attached word document and revised manuscript 

  • Yellow for reviewer 1

Yours sincerely

Authors

Reviewer 2 Report

Minor editorial corrections:

Line-14 of ABSTRACT, 'RT-PCR' needs to be spelled out

Line-23 of ABSTRACT, 'USA' needs to be spelled out

Line 52 INTRODUCTION 500.000 cases or 500,000 cases?

In INTRODUCTION and DISCUSSION:

The authors need to explain why RT-PCR was chosen over the other currently available diagnostic tools such as serological studies, RBT, iELISA. How precise and sensitive the RT-PCR over the other assays?

Expensive equipment and reagents have been used, i.e, DNA Gene All purification kit, qPCR mixes, Nanodrop, RT-PCR machines, etc. Authors need to justify the use of these expensive materials over the other cheap diagnostic tools like RBT and iELISA.

Overall, great work and presented with great clarity. Can be accepted with above minor modifications.

Author Response

Dear Reviewer,

Manuscript ID pathogens-2277585 entitled " Molecular detection and associated risk factors analysis of Brucella melitensis (B. melitensis) in aborted Sheep and Goats, in different districts of Duhok province-Iraq ".

We thank yourself  for the put into consideration of our submission and we are very happy to revise the manuscript in line with the recommendations, inputs, edits, and feedback provided.

We indicate below the changes made in text background in attached word document and revised manuscript 

  • Light green for reviewer 2

Yours sincerely

Authors

Reviewer 3 Report

The manuscript deals with the estimation of prevalence of Brucellosis in 681 blood samples of aborted small ruminants. 

Following concerns have to be addressed

1. The authors have used both RT-PCR and PCR for estimating the prevalence of Brucellosis in aborted small ruminants. The comparison between the results of these used 2 tests have to be analysed  for the estimation of prevalence of Brucellosis.

2. The management related risk factors have also to considered. Refer to the article published by Ndazigaruye et al.,2018.

3. The pathological changes and lesion scoring in different organs of sheep and goat has to be studied for the difference in tissue tropism between sheep and goats.

4. Immunohistochemical distribution of Brucella antigen and its tropism has to be compared between sheep and goats.

5. Besides molecular investigation, seroprevalence could be estimated with RBPT and ELISA. The comparison between the molecular and serological tests will strengthen your findings rather than focussing only on molecular tests. 

Author Response

Dear Reviewer, 

Manuscript ID pathogens-2277585 entitled " Molecular detection and associated risk factors analysis of Brucella melitensis (B. melitensis) in aborted Sheep and Goats, in different districts of Duhok province-Iraq ".

We thank yourself and the reviewers for the put into consideration of our submission and we are very happy to revise the manuscript in line with the recommendations, inputs, edits, and feedback provided.

We indicate below the changes made in text background.

  • Green colour for 3

Yours sincerely

Authors

Reviewer 4 Report

The article entitled “Molecular detection and associated risk factors analysis of Brucella melitensis (B. melitensis) in aborted Sheep and Goats, in different districts of Duhok province” emphasizes an important animal health-related issue. Brucella melitensis is the main causative agent of Brucellosis in sheep and goats, but sporadic cases due to B. abortus have been observed. B. melitensis causes abortion in late pregnancy, stillbirths or weak kids; following the first exposure, abortion may be in the form of a storm. Epidemiological data on Brucesllosis concerning genomic detection are generally outdated and relatively vague because studies are often imprecise or conducted in unrepresentative populations in specific geographical settings. The current study provided crucial information about the molecular detection and potential risk factors of Brucellosis in small ruminants of Duhok province. Besides the novelty of the topic, the manuscript has numerous typographical and grammatical errors.

Title: The title could be modified as “Molecular detection and associated risk factors of Brucella melitensis in aborted Sheep and Goats in Duhok province, Iraq.

Abstract: The abstract must be revised to fix all grammatical and typographical errors.

Line No. 12-13: Rewrite the sentence as “Brucellosis in sheep and goats has a significant economic and zoonotic impact on the livestock population of Duhok province, Iraq”.

Line No. 13-15:Remove the second sentence of Abstract.

Line No. 16: Add “of” after total.

Line No. 16: Replace “in” with “of”.

Line No. 17: Add “of Duhok” after districts.

Line No. 18-19: Replace “The study found” with “Results revealed an overall prevalence of 35.45% (CI=2.57) and 23.8% 18 (CI=0.44) in sheep and goats, respectively”.

Line No. 19: Add this sentence “A statistically significant (p=0.004) difference in prevalence was found between the two species”

Line No. 20: Replace “significate” with “significant”.

Line No. 23: Replace “are belong” with “were belonging”.

Line No. 25: Replace “in these areas” with “in the study regions”.

Keywords: Rephrase as: Sheep and goats; B. melitensis; risk factor; RT-PCR.

Introduction:

The introduction is too lengthy. Make it concise and add a paragraph regarding associated risk factors from the previous studies.

Line No. 30-31: Split the sentence into two.

Line No. 32: Remove the sentence “. Infections caused by Brucella are identified as bacterial brucellosis.

Line No. 34: Replace “injuries” with “disorders”.

Line No. 35: Add the complete name of B. Melitensis before its abbreviation.

Line No. 38: Add comma “,” after hosts.

Line No. 39: Write “even” before 10-100.

Line No. 43: Add a full stop (.) after worldwide. Start a new sentence after that as “Brucellosis is also classified”.

Line No. 45: Add reference here.

Line No. 48: Replace “control measures used such as” with “proper”.

Line No. 66-70: Split the sentence into two.

Material and Methods:

This section is also too lengthy. Make it concise by removing unnecessary information.

Table 1 & 2 could be presented as supplementary materials.

Section 2.4: contains a lot of grammatical errors.

Line No. 237: Replace “are belonged” with “were belonging”.

Line No. 264: Write “The” in non-italic form.

Discussion:

This section is also too lengthy. The authors are advised to avoid unnecessary explanation/ discussion of the data in this chapter. There is no need to discuss the non-significant finding, e.g., Line No. 332-339.

Conclusion:

Line No. 396: Variables including districts, sheep breed, and age of animal are not significantly associated with B. melitensis seropositivity.

The authors are advised to follow a single specific format for references.

Author Response

Dear Reviewer, 

Manuscript ID pathogens-2277585 entitled " Molecular detection and associated risk factors analysis of Brucella melitensis (B. melitensis) in aborted Sheep and Goats, in different districts of Duhok province-Iraq ".

We thank yourself and the reviewers for the put into consideration of our submission and we are very happy to revise the manuscript in line with the recommendations, inputs, edits, and feedback provided.

We indicate below the changes made in text background.

  • Gray colour for 4

Yours sincerely

Authors

Round 2

Reviewer 1 Report

I confirm that the manuscript is in a finalized form for publication, and there are no comments to be addressed by the authors in a revised version.

Reviewer 3 Report

The authors have satisfactorily revised the manuscript. 

Reviewer 4 Report

I appreciate the efforts made by the authors in improving the manuscript. They have addressed all my concerns.